# LN-Derived Fibroblastic Reticular Cells and Their Impact on T Cell Response—A Systematic Review

**DOI:** 10.3390/cells10051150

**Published:** 2021-05-10

**Authors:** Bianca O. Ferreira, Lionel F. Gamarra, Mariana P. Nucci, Fernando A. Oliveira, Gabriel N. A. Rego, Luciana Marti

**Affiliations:** 1Department of Experimental Research, Hospital Israelita Albert Einstein, São Paulo 05652-000, Brazil; bianca.oliveira@einstein.br (B.O.F.); lionel.gamarra@einstein.br (L.F.G.); mariana.nucci@hc.fm.usp.br (M.P.N.); fernando.anselmo@einstein.br (F.A.O.); gabriel.nery@einstein.br (G.N.A.R.); 2LIM44—Hospital das Clínicas da Faculdade Medicina da Universidade de São Paulo, São Paulo 05403-000, Brazil

**Keywords:** fibroblastic reticular cells, T cells, lymph nodes

## Abstract

Fibroblastic reticular cells (FRCs), usually found and isolated from the T cell zone of lymph nodes, have recently been described as much more than simple structural cells. Originally, these cells were described to form a conduit system called the “reticular fiber network” and for being responsible for transferring the lymph fluid drained from tissues through afferent lymphatic vessels to the T cell zone. However, nowadays, these cells are described as being capable of secreting several cytokines and chemokines and possessing the ability to interfere with the immune response, improving it, and also controlling lymphocyte proliferation. Here, we performed a systematic review of the several methods employed to investigate the mechanisms used by fibroblastic reticular cells to control the immune response, as well as their ability in determining the fate of T cells. We searched articles indexed and published in the last five years, between 2016 and 2020, in PubMed, Scopus, and Cochrane, following the PRISMA guidelines. We found 175 articles published in the literature using our searching strategies, but only 24 articles fulfilled our inclusion criteria and are discussed here. Other articles important in the built knowledge of FRCs were included in the introduction and discussion. The studies selected for this review used different strategies in order to access the contribution of FRCs to different mechanisms involved in the immune response: 21% evaluated viral infection in this context, 13% used a model of autoimmunity, 8% used a model of GvHD or cancer, 4% used a model of Ischemic-reperfusion injury (IRI). Another four studies just targeted a particular signaling pathway, such as MHC II expression, FRC microvesicles, FRC secretion of IL-15, FRC network, or ablation of the lysophosphatidic acid (LPA)-producing ectoenzyme autotaxin. In conclusion, our review shows the strategies used by several studies to isolate and culture fibroblastic reticular cells, the models chosen by each one, and dissects their main findings and implications in homeostasis and disease.

## 1. Introduction

Lymph node structural organization is reported to be governed by the stromal cells [1]. Fibroblast reticular cells (FRCs), a subset of the stromal cells found in the T lymphocyte region of lymph nodes (LNs) and other secondary lymphoid organs (SLOs), have been described as much more than structural cells [2].

FRCs are described to be organized in a conduit system called the “reticular fiber network”, responsible for transferring antigens from tissue to T cell zones in LNs and for controlling the conduit matrix deposition during lymph node expansion [3,4].

Their ability for cytokine and chemokine production has been demonstrated in several studies [2,5,6], and their relevant multifunctional roles and multiple subsets have been previously defined [7]. In addition, mice and human lymph node-derived FRC’s ability to react to inflammatory stimuli has been described [8,9,10].

Moreover, a few studies have implicated FRC in peripheral tolerance. Certain stromal cells can express antigens from peripheral tissues (PTA) and mediate the maintenance of peripheral tolerance through the deletion of self-reactive T cells and other mechanisms [11,12,13,14,15,16]. In addition, other cells previously known as structural cells, such as epithelium, endothelium, and fibroblasts, have also been implicated as players in the immune response [17].

However, there are several obscure points in FRC biology that need elucidation, mainly that of their dual role augmenting and, thereby, controlling the immune response. In this sense, this systematic review lists several FRC mechanisms described as controlling mechanisms of the immune response [18,19,20,21,22,23,24,25,26,27,28,29,30,31,32,33,34,35,36,37,38,39,40,41].

The articles reviewed here report on using several animal models of disease and/or genetically modified mice as tools to investigate FRCs’ effect on T cells. These articles also approach and clarify the mechanisms involved in T cell proliferation or differentiation in subsets with regulatory, effector, or memory profiles [18,19,20,21,22,23,24,25,26,27,28,29,30,31,32,33,34,35,36,37,38,39,40,41]. In addition, these articles reported the markers used to identify and isolate FRCs, as well as the methods used for these cells’ cultivation. 

Lately, FRCs’ ability for controlling the immune response and its role in several pathological conditions, such as viral infection, inflammation, metastatic cancer, and autoimmunity, are also included in this review. Consequently, we comprise here the latest updates in FRC biology, their impact on T cell fate, how they participate in diseases, and how they could be manipulated in order to ameliorate the course of certain conditions.

## 2. Materials and Methods

### 2.1. Search Strategy

As a search strategy, the only articles included were indexed and published in the last five years, including between 2016 and 2020, in PubMed, Scopus, and Cochrane, following the PRISMA guidelines [42]. The next criteria of interest selected were keywords in the following sequence: (Fibroblastic Reticular Cell OR Fibroblastic Reticular Cells) AND (Lymph Node OR Lymph Nodes) AND (T cell OR T cells). 

In addition, the following Boolean operators (DecS/MeSH) were used:

SCOPUS: (TITLE-ABS-KEY (“fibroblastic reticular cells”) OR TITLE-ABS-KEY (“fibroblastic reticular cell”)) AND (TITLE-ABS-KEY (“lymph node”) OR TITLE-ABS-KEY (“lymph nodes”)) AND (TITLE-ABS-KEY (t-cell) OR TITLE-ABS-KEY (t-cells) OR TITLE-ABS-KEY (“T cell”) OR TITLE-ABS-KEY (“T cells”) OR TITLE-ABS-KEY (“T lymphocyte”) OR TITLE-ABS-KEY (“T lymphocytes”)) AND (LIMIT-TO (DOCTYPE, “ar”)) AND (LIMIT-TO (PUBYEAR, 2020) OR LIMIT-TO (PUBYEAR, 2019) OR LIMIT-TO (PUBYEAR, 2018) OR LIMIT-TO (PUBYEAR, 2017) OR LIMIT-TO (PUBYEAR, 2016)).

PubMed: ((“T cell”[Title/Abstract]) OR (“T cells”[Title/Abstract]) OR (“T-cell”[Title/Abstract]) OR (“T-cells”[Title/Abstract]) OR (“T-lymphocyte”[Title/Abstract]) OR (“T-lymphocytes”[Title/Abstract])) AND (y_5(Filter)) AND (ffrft(Filter)) AND (fha(Filter)) AND (journalarticle(Filter)) AND (fft(Filter)) AND ((“fibroblastic reticular cells”[Title/Abstract]) OR (“fibroblastic reticular cell”[Title/Abstract]) OR (FRC[Title/Abstract]) OR (FRCs[Title/Abstract])) AND (y_5(Filter)) AND (ffrft(Filter)) AND (fha(Filter)) AND (journalarticle(Filter)) AND (fft(Filter)) AND ((“lymph node”[Title/Abstract]) OR (“lymph nodes”[Title/Abstract]) OR (“secondary organs”[Title/Abstract])) AND (y_5(Filter)) AND (ffrft(Filter)) AND (fha(Filter)) AND (journalarticle(Filter)) AND (fft(Filter)) Filters: Abstract, Free full text, Full text, Journal Article, in the last 5 years.

Cochrane: “fibroblastic reticular cells” in Title Abstract Keyword OR “fibroblastic reticular cell” in Title Abstract Keyword AND “lymph node” in Title Abstract Keyword OR “lymph nodes” in Title Abstract Keyword AND “T cell” in Title Abstract Keyword—word variations were searched.

### 2.2. Inclusion Criteria

This review included only original articles published between 2016 and 2020, and written in the English language. The following inclusion criteria were used: (i) studies in vitro and/or in vivo using fibroblastic reticular cells on a homeostatic or stimulated state and their impact on T cell function; (ii) studies of network analysis of homeostatic or stimulated fibroblastic reticular cells and their impact in T cell function.

### 2.3. Exclusion Criteria

The following exclusion criteria were used: (i) reviews, (ii) clinical articles, (iii) book chapters, (iv) protocols, (v) editorials/expert opinions, (vi) letters/communications, (vii) publications in languages other than English, (viii) indexed articles published in more than one database (duplicates), and (ix) articles that did not analyze fibroblastic reticular cells’ impact on T cell function.

### 2.4. Data Extraction, Data Collection, and Risk of Bias Assessment

In this review, all six authors (B.O.F.; F.A.O.; M.P.N.; G.N.A.R.; L.F.G.; and L.C.M.), independently and in pairs, randomly selected, revised, and evaluated the titles and abstracts of the publications identified by the search strategy in the databases cited above, and all the potentially relevant publications were retrieved in full. These same reviewers evaluated the full-text articles to decide whether the eligibility criteria were met. Discrepancies in the study selection and data extraction between the two reviewers were discussed with a third reviewer and resolved. 

B.O.F. and F.A.O. searched for the characteristics of the hosts and interventions that they received before in vitro analysis; M.P.N. and G.N.A.R. searched for the characteristics of fibroblastic reticular cell isolation and immunophenotypic labeling; M.P.N. and F.A.O. searched for the characteristics of the main type of immune cells used for analysis with fibroblastic reticular cells; B.O.F. and G.N.A.R. searched for the main characteristics of the studies used to assess the influence of fibroblastic reticular cells on the activation, expansion, or suppression of immune responses. The analysis process and table plots were carried out by the full consensus of peers, respecting the distribution above. In cases of disagreement, two senior researchers, L.F.G. and L.C.M., decided to add or subtract data. The final inclusion of the studies into the systematic review was by agreement of all reviewers. 

### 2.5. Data Analysis

All the results are described and presented using the percentage distribution for all variables analyzed in the tables.

## 3. Results

### 3.1. Selection Process of the Articles Identified According to the PRISMA Guidelines

We searched for articles indexed in the Pubmed, Scopus, and Cochrane libraries published between 2016 and 2020, and a total of 175 articles were identified in these databases. Of the 32 articles identified in Pubmed, 4 were excluded because they were reviews. After the Pubmed screening, 11 articles were not included due to the lack of analysis on FRCs’ impact on T cell function, and 17 full-text articles were assessed for eligibility. Of the 43 articles identified in Scopus, 17 were duplicated in the Pubmed database, with 26 records remaining for screening, of which 19 articles were not included due to the lack of analysis on FRCs’ impact on T cell function. A total of 7 articles from the Scopus database were assessed for eligibility. From the Cochrane database, no articles identified in the search were included in this review due to not meeting the inclusion criteria. A total of 24 studies [18,19,20,21,22,23,24,25,26,27,28,29,30,31,32,33,34,35,36,37,38,39,40,41] were included in this review and the decision stages are summarized in Figure 1.

### 3.2. Characteristics of the Host Used in the Studies Analyzing Fibroblastic Reticular Cell Function

The host characteristics in the articles included in this review, such as source, genotype, age, and gender, are described in Table 1, along with the types and times of interventions used to analyze fibroblastic reticular cell functions. The hosts used in the majority of the studies reviewed here were C57BL/6 mice (79%) [18,19,22,23,24,25,26,27,28,29,30,32,33,34,37,38,39,40,41]. There were 10 studies (53%) that used some type of genetically modified mice [21,22,25,28,30,32,33,35,38,40], of which 40% were NOD *scid gamma* (NSG) mice [21,25,33,35] and 60% were RAG deficient mice [22,28,30,32,33,35].

Of these studies, 3 [21,34,37] reported the use of mice and humans as the target hosts, and 3 studies used only humans (13%) [20,31,36]. Besides that, 4 studies (19%) used the genetic background of C57BL/6J mice [18,25,26,41], 3 (13%) used C57BL/6N [22,38,39], 2 used BALB/c2 [33,35] and C57BL/6N-Tg [38,39] (10% each). The background of NOD/ShiLtJ [21], NOR/LtJ [21], NOD.CgTg [21], ROSA26RFP [22], CD-1 IGS [25], R26R-EYFP [38], and Gbt-1.1 [40] mice were used in just 1 study each (5%). Regarding genotypes, 4 studies (19%) used knock-out or conditional knock-out mice for the cytokines IL-33 [18], IL-6 [24], IL-7 [22], and IL-17A [27]; 8 studies (38%) used knock-out or conditional knock-out mice for chemokines or reporters for CCL19 [23,25,32,38,39,40,41], and CXCL10 [40]. A total of 5 studies (24%) used knock-out or conditional knock-out mice for receptors such as IFN-αR [23], IL-23R, IL-17AR [27], TLR7 [38], CXCR3 [40], and the lysophosphatidic acid (LPA) receptors LPAR, LPAR5, and LAPR6 [41]. In addition, 5 studies also used knock-out or conditional knock-out mice for the RAG enzyme [27,28,30,33,35]. Additionally, 7 studies (33%) used knock-out or conditional knock-out mice for NOS [24,29], COX [29], MHC-II and its transcription factor CIITA [30], NOTCH2 and its Delta receptors DLL1 and DLL4 [33], and for the signaling molecules, such as ACT1 [27], MyD88 [38], and STING [40]. Finally, 3 studies (14%) used mice with specific OVA-T cell receptors [22,29,37].

The mice age range was mainly from 5 to 22 days [18,21,23,24,25,26,27,29,33,34,37,38,39,40,41], with the exception of the studies of Masters [28] and Dubrot [30], which used aged mice (older than 12 months). The animals’ genders were reported in only 54% of the studies, of which 46% used only males [18,19,24,25,26,28], 15% used only females [21,34], and 39% used both [27,33,35,36,40]. Around 71% of the studies [18,19,23,24,25,27,28,29,30,32,33,34,37,38,39,40,41] used some type of host intervention, such as a diphtheria toxin [19,25,32,39] or infection, using different types of virus or bacteria (LCMV [18,23,24], WE [18], PR8-GP33-41 [24], influenza [24,28], HSV-1 [40], mouse hepatitis virus A59, and Citrobacter rodentium [38]), or strategies that simulate viral infection, such as poly (I:C) [29,33], associated with irradiation or only irradiation [19], as well as immunization with OT-1 T cells with OVA [24,27], the use of drugs such as tamoxifen [18,30], or immunosupressors such as FTY720 [30], among others, with varied times of application.

### 3.3. General Immunophenotypic Characteristics of Fibroblastic Reticular Cells

The main characteristics of FRCs, lymph node (LN) digestion processes, and techniques used for their isolation are described in Table 2.

#### 3.3.1. Lymph Node Characteristics

Among the 24 studies selected for this review, 12 (50%) used FRCs derived from peripheral LNs (axillary, skin-draining, cervical, inguinal, popliteal, kidney, mandibular, mediastinum, pancreatic) [18,21,23,25,29,30,32,33,34,37,39,40], 7 (29%) used a combination of peripheral and mesenteric LNs [19,20,22,26,27,28,41], 3 (13%) studies used only mesenteric LNs [35,36,38], and 2 studies did not disclose the LN source [24,31].

#### 3.3.2. Tissue Disaggregation Type

Regarding the type of tissue disaggregation used in the studies, most of them (83%) describe enzymatic digestion [18,19,20,21,22,23,24,27,28,29,31,32,33,34,35,37,38,39], 2 studies (8%) performed mechanical tissue disruption [36,40], and another 2 used models of topological analyses [25,26], all described in Table 2. For the enzymatic digestion process, 38% of the studies used a combination of solutions, including Collagenase P, Dispase (I or II) and DNase I, [21,22,24,31,32,35,37,40,41]. In addition to the previous solution, the study of Knop et al. used Latrunculin B [22]. Another 8 studies (33%) used only a combination of Collagenase (IV, D, F, or P) and DNase I [23,29,30,33,34,38,39], and the study of Aparicio-Domingo et al. included CaCl2 in the final solution [18]. The work of Dertschnig and Eom used DNase, Liberase and DNase I, or Liberase DH, respectively [19,20]. Masters et al. were the only ones to use the combination of Liberase TL and Benzonuclease [28].

#### 3.3.3. Culture Media Details

Of the culture media used for FRC cultivation, RPMI was the most popular medium and was used in 38% of the studies, even though there were differences in its supplementation. A total of 9 studies used 2 to 10% of fetal bovine serum (FBS) for supplementation [20,22,28,34,35,36,38,40,41]. In addition, 5 studies (21%) used DMEM as a culture medium, supplemented with 2% of fetal calf serum (FCS) or 2 to 10% of FBS [18,29,32,33,37]. Another 2 studies mentioned the use of RPMI without supplementation [23,27], and Novkovic’s study used RPMI supplemented with 2% of FCS [39]. Brown and Knoblich’s studies used α-MEM [24,31]; Dubrot, HBSS [30], Gonzalez did not disclose the culture medium used [21], and Dertschnig et al. did not cultivate the FRCs [19].

#### 3.3.4. FRC Characterization and Selection

Most studies—18 of the 24 (75%)—used the classical markers for FRC characterization, using antibodies to determine a population CD45-negative, CD31-negative, and podoplanin (PDPN or gp38)-positive [18,19,20,21,22,25,27,28,29,30,31,32,33,34,35,36,37,38]. Some studies (13%) did not disclose the FRC characterization [23,40]; Knoblich et al. used CD45 and PDPN [31]. In regard to cell separation methods, the technique most used was cell sorting, using FACS (46%) [18,19,21,22,24,30,33,35,37,38,41], even though 6 studies did not report the used strategy [20,23,31,32,36,40]. Another 4 studies did not use cell sorting in their protocol [25,26,35,38]. In addition, 3 other studies used microbeads for cell separation [27,28,29]. Lastly, most studies (46%) did not report the purity resulting from their cell selection [19,21,23,30,32,33,36,38,40,41]. Only 9 studies (38%) reported a purity between 73–100% [18,22,24,27,28,29,31,35,37].

### 3.4. Immunophenotypic Characteristics of Immune Cells Commonly Used in Studies with Fibroblastic Reticular Cells

#### 3.4.1. Immune Cell Origins

Secondary lymphoid organs (SLO) comprise a variety of immune cells and non-immune cells; in the studies selected for this revision, the non-immune cells were FRCs. In this context, these studies verify the relation between FRCs and immune cells found in SLOs. Most studies used LNs (42%) [18,19,20,24,26,27,34,35,36,37] or spleens (29%) [22,23,25,29,30,39,41] as an immune cell source, 13% of the studies used, besides LNs, peripheral blood, tonsils, and Peyer’s Patches cells [28,31,38]. Masters et al. [28] used peripheral blood cells in addition to LNs, Chung and Gonzalez used spleens instead of LNs [21,33], and Royer used several SLOs [40], all described in Table 3.

#### 3.4.2. Immune Cell Types

The cells used in combination with FRCs in most studies (29%) were total T lymphocytes [19,29,31,32,34,35,37], followed by only T CD8 lymphocytes (20%) [18,21,24,28,40]; 2 studies (8%) used T CD4 lymphocytes [25,36]; 2 studies (8%) used a combination of T cells and B cells [27,41]; 2 studies (8%) did not evaluate the presence of immune cells [20,26]. Moreover, Knop et al. used T cells and NK cells [22]; Perez-Shibayama et al. used T cell subsets and exhaustion makers [23]; Dubrot et al. used a combination of T cells, B cells, Treg, and DCs [30]; Chung et al. used the same cells as Dubrot et al. but included FDCs in their study [33]; Gil-Cruz et al. used T cells, B cells, NK cells, Treg, and ILCs [38]; and Novkovic et al. used DCs and T cells [39], each study representing 4% of the total.

#### 3.4.3. Immune Cell Selection

Regarding the separation techniques of immune cells, 4 studies (17%) used Pan T isolation by negative selection [30,31,32,37], 4 studies did not perform selection [18,23,25,29], and 4 studies did not report the separation technique [27,34,38,39]. Another 3 studies (13%) used CD8 positive selection [22,24,40], 2 studies (8%) used CD8 isolation by negative selection [21,28], 2 studies used CD4 naïve T cell negative selection [36,41], and for 2 other studies, this method was not applicable [20,26]. Dertschnig et al. used CD3 negative selection, followed by CD4 and CD8α positive selection [19]; Chung et al. used T cell Thy.1 selection [33]; and Pazstoi et al. used CD4 positive selection [35], each study representing 4% of the total.

#### 3.4.4. Immune Cell Culture

In regards to the media used for immune cells, 38% of the studies used RPMI as the same medium used to culture the FRCS [22,23,29,36,37,38,39,40,41]. A total of 5 studies (21%) used RPMI supplemented with 10% of FBS [22,23,37,40,41]. Another 4 studies used RPMI supplemented with 2% to 10% of FBS [29,36,38,39]. Aparicio-Domingo et al. used DMEM with 2% of FCS [18]; Brown et al. used both RPMI and α-MEM [24], while Pazstoi used X-VIVO [35]. Another 9 studies (38%) did not disclose the medium used [19,21,25,27,28,30,31,32,34].

### 3.5. Studies Used to Assess the Influence of Fibroblastic Reticular Cells on Immune Response

The studies used different strategies in order to access the contribution of FRCs to different mechanisms involved in the immune response (Table 4). A total of 5 studies (21%) evaluated viral infection in this context: 2 of them used LCMV [18,23], 1 used influenza [24,28], 1 used HSV-1 [40], and 1 used both LCMV and influenza [18]. Another 4 studies (17%) used inflammation as a model for targeting the COX:PGE_2_ pathway [29,31,36,37]. Another 3 studies (13%) used a model of autoimmunity, including Type 1 diabetes [21], glomerulonephritis [25], and experimental autoimmune encephalomyelitis (EAE) [27]. Another 2 studies (8%) used a model for GvHD [19,33], 2 studies used cancer (8%) as models of study—1 melanoma [20] and 1 tumor-draining lymph nodes [34]. One study (4%) used a model of ischemic-reperfusion injury (IRI) [32]. Another 4 studies targeted just one particular pathway, such as MHC II expression [30], FRC microvesicles [35], FRC secretion of IL-15 [38], the FRC network [39], or the ablation of the LPA-producing ectoenzyme autotaxin in FRCs [41].

Most studies, 21 (87%), used flow cytometry in order to separate and/or evaluate their cell populations and results [18,19,20,21,22,23,24,25,27,28,29,30,31,32,33,34,35,36,37,38,40,41]. Another 6 studies (25%) used RNA sequencing or gene expression in order to have more broad information about their models [18,19,20,24,25,31,35]. A total of 4 studies (16%) used, in addition to FC, intravital and/or confocal microscopy in order to complement their results [19,21,32,41]. In addition, 2 studies (8%) were complemented with siRNA and Western blotting [27,37]. Moreover, 2 studies [8%] only used intravital and/or confocal or electronic microscopy to evaluate their models [26,39].

#### 3.5.1. The Role of FRCs in the Immune Response Varied According to the Model Studied

##### Anti-Viral Response

In studies related to the anti-viral response, Aparicio-Domingo et al., in an LCMV study, concluded that FRCs displayed a stimulatory role, being an important source of IL-33 in the lymph node and vital for driving acute and chronic antiviral T cell responses [18], while Perez-Shibayama et al., who also used LCMV as a model and found a regulatory role for FRCs, showed an IFN-α-signaling dependent shift of FRCs toward an immunoregulatory state, reducing exhaustive CD8 T lymphocyte activation [23]. Yet, in an anti-viral response for influenza, Brown et al. [24] showed that FRCs have a role beyond a regulatory one in reducing T cell expansion—they also outline the fate and function of CD8 T lymphocytes through their IL-6 production. Moreover, Masters et al. reported that after aging-related changes, FRCs have their impact altered on the initiation of the immune response to influenza infection, and this contributes to delayed T lymphocytes responses to this virus [28]. Finally, Royer et al. proposed, in their study, that HSV-1-infected lymph nodes can cause pathological alterations in FRC conduit systems, resulting in fewer HSV-specific CD8 T lymphocytes in circulation, and a diminished anti-viral response to this virus [40].

##### Inflammatory Response

Regarding inflammation and the COX:PGE_2_ pathway, Schaeuble et al. found that FRCs constitutively express high levels of COX2 and its product PGE_2_, thereby identified as a mechanism of T lymphocyte proliferation control [29] Knoblich et al. also demonstrated that FRCs control T cell proliferation and modulate their differentiation [31]. Yu, M. et al. also agree that a hyperactivated COX-2/PGE_2_ pathway in FRCs is a mechanism that maintains peripheral T cell tolerance [37]. Valencia et al. discussed the differences between mice and humans regarding COX inflammatory pathways, and concluded that human and murine FRCs would regulate T lymphocyte responses using different mechanisms, arguing that in humans, IDO would play a more important role than iNOS/NO [36].

##### Autoimmunity

Further, an autoimmunity FRC network seems to play an important role. Gonzalez et al., using a type 1 diabetes model, found that FRCs modulate their interactions with autoreactive T lymphocytes by remodeling their reticular network in LNs, and podoplanin plays a central role in this alteration [21]. Kasinath et al., using glomerulonephritis (GN), showed that the removal of a kidney-draining lymph node, the depletion of fibroblastic reticular cells, or treatment with anti-podoplanin antibodies all resulted in a reduction of kidney injury in GN [25]. Finally, Majumder et al., in EAE, showed that Th17 differentiation in LNs leads to IL-17 signaling to FRCs and an impact on LN stromal organization by promoting FRC activation through a switch on their phenotype from quiescence to highly metabolic [27].

##### GvHD

Moreover, in graft versus host disease (GvHD), FRCs’ abilities for peripheral tissue antigen (PTA) presentation and NOTCH signaling have shown to be important features for the aggravation and maintenance of this state. Dertschnig et al. showed that the loss of PTA presentation by FRCs during GVHD leads to permanent damage in their networks in lymphoid tissues [19], and Chung et al. showed that FRC-delivered NOTCH signals to donor alloreactive T cells help to program the pathogenicity of these T cells [33].

##### Metastatic Cancer

In metastatic cancer, FRCs appear to be regulated by the tumor cells. Eom et al. showed in melanoma that FRCs in tumor-infiltrated LNs may favor cancer invasion and progression through the secretion of soluble factors [20], and Gao et al. also showed in tumor-infiltrated LNs a decrease in FRCs and IL-7 secretion, leading to a declined number of T lymphocytes in LNs [34].

##### Renal Injury

As seen by Kasinath et al. in GN [25], Maaraouf et al. using ischemic reperfusion injury (IRI) also confirmed that depletion of FRCs reduced T cell activation in the kidney LNs and ameliorated renal injury in acute IRI [32].

##### Specific Signaling Pathways

Regarding pathway investigation, Knop et al. described the essential role of IL-7 derived from FRCs for central memory T cell survival [16]; Dubrot et al. showed a mechanism of T lymphocyte proliferation inhibition by the induced expression of MHC II [30]; Pazstoi et al. described that FRCs contribute to peripheral tolerance by fostering de novo Treg induction by MVEs carrying high levels of TGF-β [35]; Gil-Cruz et al. commented on the role of FRCs on innate lymphocytes ILC1 and NK through IL-15 secretion [38]; Novkovic confirmed that the physical scaffold of LNs formed by the FRC network is critical for the maintenance and functionality of LNs [39]; Takeda et al. demonstrated the role of LPA derived from FRCs in T lymphocyte motility [41].

## 4. Discussion

FRC is a specific subset of stromal cells present in the lymph node, and they are precisely located in the T cell zone. There are other stromal cell subsets in lymph nodes, described as double-negative cells, follicular dendritic cells, blood endothelial cells, lymphatic endothelial cells, and others that are not discussed in this review [43].

The results of this review firstly show the characteristics of the host type used for analyzing FRC function. In addition, strategies used by them in order to achieve their target objectives, including model characteristics, such as source, genotype, age, and gender, are described in Table 1. The main characteristics of FRCs, their origin, as well the lymph node (LN) digestion process, and techniques used for their isolation are described in Table 2. The immune cell sources, as well as their characteristics, are described in Table 3. All these variations between the models studied, cell origins, and characterization, sometimes lead to different conclusions, making the comparison between studies difficult or conflicting, such as the role of FRCs in T cell proliferation, sometimes described as stimulators and, at other times, as limiting. Next, we assembled the studies with the same subject (Table 4) and compared them, trying to show the differences and, more importantly, comparisons between the achieved results (Figure 2).

The first scenario discussed was on viral infection (Figure 2A). Aparicio-Domingo et al., in an LCMV study, concluded that FRCs displayed a stimulatory role, being a main source of IL-33 in the lymph node and crucial for leading to acute and chronic antiviral T cell responses. They also showed that FRCs mainly act on CD8 T lymphocytes by signaling via ST2 expressed by these T cells [18]. Severino et al. demonstrated previously, in 2017, the increased IL-33 gene expression in human FRCs after treatment with IFN-γ or IL-1β and TNF-α. These cytokines are usually released during a course of an immune response, supporting the Aparicio-Domingo et al. findings that FRCs are the main source for IL-33 [9].

Perez-Shibayama et al., using the LCMV model like Aparicio-Domingo et al., commented that FRCs contributed to an immunostimulatory state to prevent virus replication and spread. However, they also found a regulatory role of FRCs, showing an IFN-α-signaling dependent shift of FRCs toward an immunoregulatory state, reducing exhaustive CD8 T lymphocyte activation. They claim that type 1 IFN-mediated control of LCMV replication in FRCs is one of the major factors that determine the quality of the antiviral CD8+ T cell response [23]. In agreement, Talemi and Hofer sustain the idea that interferons delay the viral spread in infection, acting as sentinels, warning uninfected cells, and also are negative feedback regulators acting at a single-cell level [44].

Regarding the anti-viral response for influenza and LCMV, Brown et al. [24] showed that FRCs function is more than controlling T cell expansion. FRCs also outline the fate and function of CD8 T lymphocytes through their IL-6 production, and CD8 T cells exposed to both FCRs and IL-6 are driven to a memory phenotype. In addition, CD8 T cells cultivated in the presence of FRCs are more persistent during a viral infection than CD8 T cells stimulated without FRC presence [24]. Moreover, the pleiotropic function and the importance of IL-6 were reported before, supporting that this cytokine, in certain environments, could be an important player for guiding the immune response [45]. Next, Masters et al., reported that after aging-related changes, FRCs have an altered impact on the beginning of the immune response to influenza infection, consequently contributing to delayed T lymphocytes responses to this virus [28]. Moreover, their findings on the importance of homeostatic chemokines for the success of the anti-viral response are also supported by Chai et al., who previously reported on the importance of these chemokines secreted by FRCs to the immune response against virus infection, and by Thompson et al., who also reported on the role of the lymph node in aging mice and its negative impact on T cells [46,47]. Lastly, Royer et al. proposed that HSV-1 in lymph nodes can cause pathological alterations in the FRC conduit system, resulting in fewer HSV-specific CD8 T lymphocytes in circulation, and a diminished anti-viral response to this virus. In addition, they claim that immunodeficiency can occur as a secondary outcome of FRC alterations to SLOs [40]. Their results are supported by other models that impair T cell responses due to virus-associated damage to FRCs [48,49].

Concerning inflammation (Figure 2B) and the COX/PGE_2_ pathway, which converts arachidonic acid in several prostanoids via the enzymes COX1 and COX2, FRCs have been proposed to play dual roles by either promoting or inhibiting adaptive immunity [50,51], similar to myeloid and T cells. Schaeuble et al.’s experiments revealed that FRCs can control T cell responses, independently of other cells, by two pathways that lead to NO release, clarifying that one pathway is activated via the sensing of IFN-y by FRCs, which is activated only by strong T cell responses, and another pathway is mediated by COX2-dependent synthesis of PGE_2_, which signals via EP1 and EP2 during both weak and strong T cells responses [29]. Knoblich et al. also demonstrated that FRCs control T cell proliferation and modulate their differentiation [31]. Knoblich et al. included even more mechanisms that control T cell proliferation besides IFN-y and PGE_2_, which, in human cells, do not release NO, but instead activate IDO; they point to TGF-β and the adenosine 2A receptor (A2AR) as other signaling pathways affecting T cell proliferation. They also demonstrated that human FRCs affect the fate of naïve T cells, diminishing their differentiation into central memory while enhancing effector and effector memory phenotypes [31]. Yu, M. et al. support these findings with their previous study on the animal model and in vitro assays, confirming that hyperactive COX-2/PGE_2_ pathways in FRCs are a mechanism that maintains peripheral T cell tolerance [37]. In addition, Valencia et al. demonstrated the differences between mice and humans regarding COX inflammatory pathways, and concluded that human and murine FRCs would regulate T lymphocytes responses using different mechanisms, and arguing that, in humans, IDO would play a more important role than iNOS/NO [36].

Further, in autoimmunity, the FRC network seems to play an important role (Figure 2C). Gonzalez et al., using a type 1 diabetes (T1D) model and a 3D system of culture, found that in T1D FRCs, the reticular network organization was altered, displayed larger pores, and had a lower expression of podoplanin compared to a control animal or control culture system. They also demonstrated a reduced expression of PTAs and T1D antigens in T1D FRCs. Consequently, FRCs modulated their interactions with autoreactive T lymphocytes by remodeling their reticular network in LNs; PTAs and podoplanin played a central role and their alterations may favor T1D [21]. These findings are supported by a previous study from the same group that investigated alterations in pancreatic lymph nodes from humans and mice [52]. Kasinath et al. studied crescentic glomerulonephritis (GN), an autoimmune inflammatory condition characterized by the rapid deterioration of kidney function. They investigated the role of fibroblastic reticular cells residing in the stromal compartment of the kidney lymph node in this model. They observed that FRCs are fundamental to the propagation of the immune response in nephrotoxic serum nephritis. Following GN development, they observed an increase in effector memory and Th17 cells in the kidney LN. In addition, they observed that the removal of the kidney lymph node, a depletion of fibroblastic reticular cells, and treatment with anti-podoplanin antibodies each resulted in a reduction of kidney injury [25]. Majumder et al. studied the EAE model, and they also showed Th17 differentiation in LNs and that the signaling in the receptor for IL-17 in FRCs is related to collagen deposition in LNs. This work suggests that Th17 cells promote ECM deposition in inflamed LNs through FRCs-IL-17 signaling, independently of LN size or hypercellularity. As a consequence of Th17 in LN, the released IL-17 signals in FRCs impact LN stromal organization, leading to FRC activation by changing their phenotype from quiescence to highly metabolic. Moreover, the absence of IL-17 signaling in FRCs does not lead to immune failings but does cause impaired B cell responses, due to the reduced availability of BAFF, which is critical for the germinal center formation and maintenance [27].

In metastatic cancer (Figure 2D), FRCs appear to be regulated by the tumor cells. Eom et al. showed in human melanoma that FRCs in tumor-infiltrated LNs may favor cancer invasion and progression through secretion of soluble factors, alterations in the lymph node structure, and by promoting pathological conditions such as fibrosis [20]. Gao et al. also showed in tumor-infiltrated LNs a decrease in FRCs and IL-7 secretion, leading to a declined number and diminished function of T cells in LNs [34].

In homeostasis, as displayed in Figure 2E, FRCs played an important role in secreting homeostatic chemokines, promoting the meeting between T cells and dendritic cells on the T cell zone, and also by secreting IL-7, an essential cytokine involved in T cell effector memory differentiation [2,14,28].

Furthermore, in graft versus host disease (GvHD) FRCs’ ability for peripheral tissue antigen (PTA) presentation and NOTCH signaling have been shown to be important features in the aggravation and maintenance of the GvHD state (Figure 2F). Dertschnig et al. showed that the loss of PTA presentation by FRCs during GVHD leads to permanent damage in their networks in lymphoid tissues, compromising peripheral tolerance. In addition, they demonstrated that not only the disruption of FCRs occurs during GvHD but also the capacity for the regeneration of this network is impaired, different to what was found for viral infection, where the damage occurs, but after viral clearance, the network is restored [19]. Chung et al. showed that FRC-delivered NOTCH signals through the ligands DLL1 and DLL4 to donor alloreactive T cells help to program the pathogenicity of these T cells. Moreover, they demonstrated that the early use of neutralizing antibodies against DDL1 and DDL4 abrogated GvHD [33].

As seen by Kasinath et al. in GN [25], Maaraouf et al., using ischemic reperfusion injury (IRI) with multiple IRI [32], reported that kidney LNs (KLNs) cause excessive deposition of ECM fibers containing fibronectin and collagen, which leads to local fibrosis, similar to kidney fibrosis. They confirmed that depletion of FRCs reduced T cell activation in the KLNs and ameliorated renal injury in acute IRI [25,32].

Regarding pathway investigation, Knop et al. demonstrated that FRC-derived IL-7 plays an essential role in maintaining central memory T cells, but is dispensable for naïve T cell survival [22]. Dubrot et al. showed a mechanism of T lymphocyte proliferation inhibition by the IFN-y-induced expression of MHC II [30]. In addition, they demonstrated that the deletion of MHC II in LN stromal cells in vivo leads to diminished Treg frequency and functions, and, at the same time, enhances effector cell differentiation, further leading to T cell tissue infiltration and the subsequent development of T cell-mediated autoimmunity [30]. Pazstoi et al. used the stromal compartment of gut-draining LNs to demonstrate that FRCs own the tolerogenic capacity that controls T cells. They also demonstrated that mesenteric LNs (mLNs) are more capable of inducing [35] Treg profiles than the peripheral ones. Likewise, they demonstrated that FRCs contribute to peripheral tolerance by developing de novo Treg by releasing microvesicles (MVEs), which carry high levels of TGF-β [35]. Gil-Cruz et al. also used mLNs and Peyer’s patches (PP) as the targets of their study and identified that an antiviral response driven by ILC1 and NK was regulated by the FRCs’ limiting provision of IL-15 [38]. This mechanism control seems to be activated by TLR7 and/or IL-1β, and its control is designated by the MyD88-dependent pathway [38]. Novkovic confirmed that the physical scaffold of LNs formed by the FRC network is critical for the maintenance and functionality of LNs [39], and Takeda et al. demonstrated the role of LPA derived from FRCs in T cell motility [41]. Kelch et al. demonstrated, by imaging, the conduit density in the deep and superficial T cell zone, concluding that although T cells within the superficial zone stay in constant contact with FRCs, and in the deep T cell zone, there is a gap that does not guarantee simultaneous contact for all T cells in this region [26].

In summary, FRCs in homeostasis plays an important role in secreting homeostatic chemokines and IL-7, which are essential for the immune response initiation and for T cell effector memory phenotype differentiation. In a viral setting, FRCs are the main source of IL-33, playing a regulatory role by diminishing the T cell exhaustion, and acting on T cell fate through IL-6 secretion. In this same setting, aging FRCs have a negative impact on T cells. In inflammation, FRCs have been proposed to play a dual role by either promoting or inhibiting adaptive immunity. The main mechanisms behind inflammation are related to IFN-y and PGE_2_-signaling that, in murine cells, release NO and, in humans, activate IDO. In autoimmunity, the reticular network organization was altered, displayed lower expression of PTAs and podoplanin, and, in this context, IL-17 signaling impacted LN stromal organization, leading to highly metabolically activated FRCs. In metastatic cancer, FRCs appear to be regulated by the tumor cells decreasing IL-7 secretion and enhancing other soluble factors, causing alterations in the lymph node structure, such as fibrosis. In GvHD, the loss of PTA presentation by FRCs leads to permanent damage in their networks, compromising peripheral tolerance.

## 5. Conclusions

These studies reviewed here contributed to the development of deep basic knowledge on immune regulation by FRCs, which seems to be the key to achieving innovative treatments for immune-related diseases and immune-mediated deficiencies. Moreover, these studies have made advances in unveiling cells and molecules that are able to regulate T cell activation during conditions such as inflammation, viral infection, metastatic cancer, autoimmunity, and GvHD, besides dissecting the pathways in FRCs and the lymph node paracortex structure.

## Figures and Tables

**Figure 1 cells-10-01150-f001:**
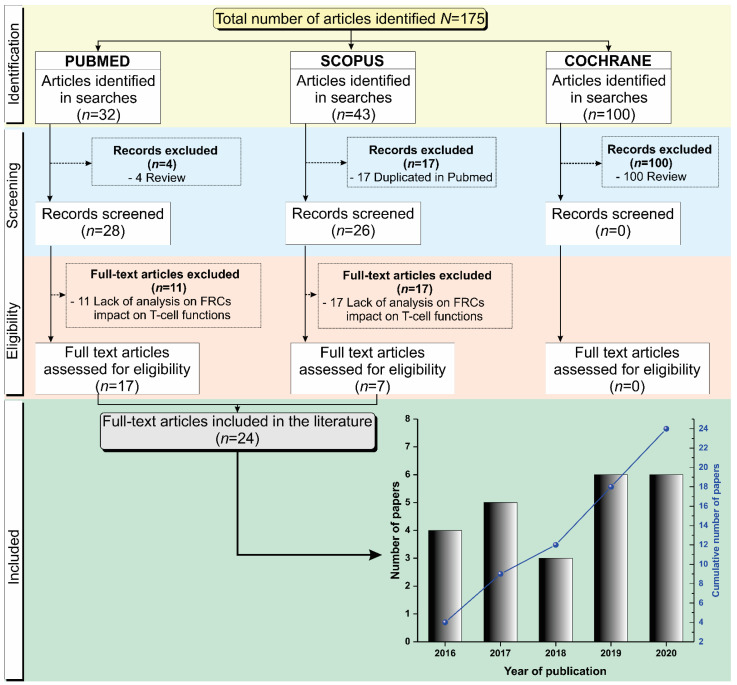
Schematic representation of the article screening process for their inclusion in this systematic review following the PRISMA guidelines. The histogram on the temporal distribution of the 24 articles included in this systematic review, by year of publication.

**Figure 2 cells-10-01150-f002:**
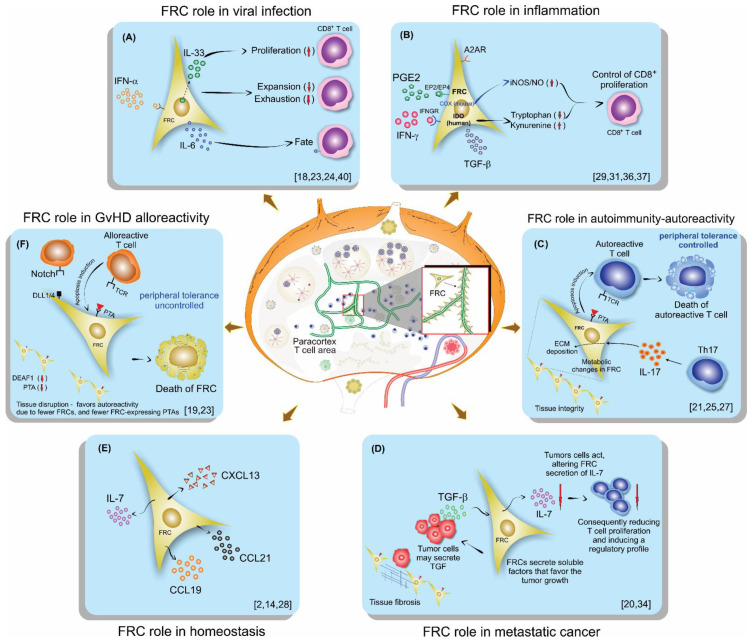
Schematic illustration of lymph nodes, FRC localization, and their role on lymphocytes in different scenarios of the immune response: (**A**) Viral infection, (**B**) Inflammation, (**C**) Autoimmunity, (**D**) Metastatic cancer, (**E**) Homeostasis, (**F**) GvHD.

**Table 1 cells-10-01150-t001:** Characteristics of hosts and the interventions that they received before in vitro analysis.

Ref.	Year	Host				Interventions	
		Source	Genotype	Age (Weeks)	Gender	Type	Time (Days)
Aparicio-Domingo et al. [18]	2020	Mice C57BL/6J	IL-33^gfp/gfp^; IL-33^gfp/+^	7–19	M	LCMV clone 13 and WE virus; tamoxifen	Single dose; 6 (3/week)
Dertschnig et al. [19]	2020	Mice C57BL/6	Female to male bone marrow transplant model (BMT), T cell-depleted, plus transgenic TCR-CD8 *MataHari* (Mh)	NR	M	Dexamethasone; DT; Gy irradiation	3; 4
Eom et al. [20]	2020	Human	Metastatic melanoma and surgery	NA	NA	NA	NA
Gonzalez et al. [21]	2020	Mice (NOD/ShiLtJ, NOR/LtJ, and NOD.CgTg); Human	Type 1 diabetes	12	F	NA	NA
Knop et al. [22]	2020	Mice C57BL/6N and ROSA26RFP	IL-7^−/−^, PGK-Cre, FLPO, RAG1^−/−^, Thy1.1+ OT-I	NR	NR	NA	NA
Perez-Shibayama et al. [23]	2020	Mice C57BL/6	CCL19-Cre IFNAR^fl/fl^	8–10	NR	LCMV Armstrong	NR
Brown et al. [24]	2019	Mice C57BL/6	IL-6^−/−^; NOS2^−/−^	5–12	M	PR8-GP33-41, LCMV, influenza, OT-1 T cells with OVA	NR
Kasinath et al. [25]	2019	Mice CD-1 IGS or C57BL/6 or C57BL/6J	CCL19-Cre iDTR	8–10	M	Nephrotoxic serum (NTS); DT	3
Kelch et al. [26]	2019	Mice C57BL/6J	NA	9–22	M	NA	NA
Majumder et al. [27]	2019	Mice C57BL/6	IL-17A^−/−^; IL-17RA^fl/fl^; OT-II, ACT1^−/−^; CCL19-Cre; IL23R^−/−^; Regnase1^+/-^	6–12	M-F	MOG with Mycobacterium tuberculosis, pertussis toxin on/OT-II CD4+ T cells with OVA/DSS	2
Masters et al. [28]	2019	Mice C57BL/6	RAG^−/−^; CD45.1	2–4 m and 19–21 m	M	Influenza	NR
Schaeuble et al. [29]	2019	Mice C57BL/6	NOS2^−/−^; OT-1; COX2^−/−^, COX2^Δ^^CCL19Cre^, and ROSA26-EYFP^CCL19Cre^	≥6	NR	OVA and poly (I:C)	4
Dubrot et al. [30]	2018	Mice C57BL/6	CIITA^−/−^; pIV^−/−^; K14 TGP IVKO; RAG2^−/−^ PROX-1-Cre MHC-II^fl^	>12m	NR	Tamoxifen; IFN-γ and FTY720	4 (Twice/day); 6
Knoblich et al. [31]	2018	Human	Cadaveric donors	NA	NA	NA	NA
Maaraouf et al. [32]	2018	Mice C57BL/6	CCL19-Cre; iDTR; RAG1^−/−^	NR	NR	DT; LTβr-Ig	1; 2
Chung et al. [33]	2017	Mice BALB/c or C57BL/6	TgMx1-Cre; DLL1^fl/fl^; DLL4^fl/fl^; NOTCH2^fl/fl^; RAG1^−/−^	6–10 or 8–12	M-F	poly (I:C)/8.5-9 Gy; poly (I:C)/6 Gy irradiation	0.16; 0.12
Gao et al. [34]	2017	Mice C57BL/6 and Human	Colon cancer	6	F	Lewis Long carcinoma cells	NA
Pazstoi et al. [35]	2017	Mice BALB/c	FOXP3hCD2xRAG2^−/−^ xD011.10	NR	M-F	NA	NA
Valencia et al. [36]	2017	Human	Brain-dead organ donors	NA	M-F	NA	NA
Yu, M. et al. [37]	2017	Mice C57BL/6 and Human	PTGS2^Y385F/Y385F^; OVA-specific CD8 (OT-I); CD4 (OT-II)	4–6	NR	DC-vaccine	1.5
Gil-Cruz et al. [38]	2016	Mice C57BL/6N or C57BL/6N-Tg or R26R-EYFP	Myd88^−/−^; TLR7^−/−^; CCL19-Cre	8–10	NR	MHV A59; Citrobacter rodentium	12; 6
Novkovic et al. [39]	2016	Mice C57BL/6N or C57BL/6N-Tg	CCL19-Cre; iDTR	6–9	NR	DT	3 and 5
Royer et al. [40]	2016	Mice C57BL/6 or Gbt-1.1	CXCL10^−/−^; CXCR3^−/−^; STING^−/−^; CD18^−/−^	6–12	M-F	HSV-1	NR
Takeda et al. [41]	2016	Mice C57BL/6J	LPAR2^−/−^; ENPP2-flox, CCCL19-Cre, LPAR5^−/−^; LPAR6^−/−^	8–12	NR	CD4+ T cells labeled with CMTMR; LTβR-Fc	0.6; 1.04; 28

Abbreviations—Ref.: reference; NR: not reported; NA: not applicable; M: male; F: female; DT: diphtheria toxin; NTS: nephrotoxic serum; DC: dendritic cells; LCMV: lymphocytic choriomeningitis virus; WE: lymphocytic choriomeningitis virus strain WE; MOG: myelin oligodendrocyte glycoprotein; OVA: ovalbumin; HSV-1: herpes simplex virus 1; DSS: dextran sodium sulfate colitis; MHV: mouse hepatitis virus; CMTMR: cell tracker; LTβR: lymphotoxin-β receptor; FTY720: immunomodulator, IL: interleukin, TCR: T cell receptor, RAG1: recombination activating gene 1; IFNAR: interferon-α/β receptor; NOS2: nitric oxide synthase 2; CCL19: chemokine (C-C motif) ligand 19; PGK: phosphoglycerate kinase 1; FLPO: is an artificial derivative of the recombinase encoded by the saccharomyces cerevisiae 2μ plasmid; Thy1.1: thymus cell antigen 1.1; OT-I: ovalbumin TCR-I; OT-II: ovalbumin TCR-II; iDTR: inducible diphtheria toxin receptor; ACT1: adaptor for IL-17 receptors; COX2: cyclooxygenase-2; EYFP: enhanced yellow fluorescent protein; CIITA: class II transactivator factor; pIV-promoter IV, MHC-II: major histocompatibility complex class II; PROX1- prospero homeobox 1; DLL: delta; FOXP3: forkhead box P3; PTGS2: prostaglandin endoperoxide synthase 2; Myd88: myeloid differentiation primary response 88; TLR7: toll-like receptor 7; CXCL10: C-X-C motif chemokine ligand 10; CXCR3: C-X-C motif chemokine receptor 3: STING: stimulator of interferon response; LPAR: lysophosphatidic acid receptors; ENPP2: ectonucleotide pyrophosphatase/phosphodiesterase 2; IFN-γ: interferon gamma; Gy: gray.

**Table 2 cells-10-01150-t002:** Characteristics of fibroblastic reticular cells isolation and their immunophenotype.

Ref.	Lymph Node Region	Digestion Type	Digestion Solution	FRC Culture Medium + Supplement	FRC Immunophenotypic Characterization	Technique for Cell Separation	Purity (%)
Aparicio-Domingo et al. [18]	Axillary; brachial; inguinal	Enzymatic	Collagenase IV; DNase I; CaCl_2_	DMEM (2% FCS)	CD45; CD31; PDPN	Cell sorting	>94
Dertschnig et al. [19]	Peripheral; mesenteric	Enzymatic	DNase; Liberase	NC	CD45; CD31; PDPN	Cell sorting	NR
Eom et al. [20]	Axillary; inguinal; cervical; mesenteric; mediastinum	Enzymatic	DNase I; Liberase DH	RPMI-1640	CD45, CD31, PDPN	NR	NR
Gonzalez et al. [21]	Skin-draining (brachial; axillary; inguinal); Pancreatic	Enzymatic	Collagenase P; DNase I; Dispase II	NR	CD45; CD31; PDPN	Cell sorting	NR
Knop et al. [22]	Peripheral; mesenteric	Enzymatic	Collagenase P; Dispase II; DNase I; Latrunculin B	RPMI-1640	CD45; CD31; PDPN	Cell sorting	>73.3
Perez-Shibayama et al. [23]	Inguinal	Enzymatic	Collagenase F; DNase I	RPMI	NR	NR	NR
Brown et al. [24]	NR	Enzymatic	Collagenase P; DNase I; Dispase	α-MEM	CD45; CD31; PDPN	Cell sorting	>95
Kasinath et al. [25]	Kidney	NA	NA	NA	NA	NA	NA
Kelch et al. [26]	Popliteal; mesenteric; Inguinal	NA	NA	NA	NA	NA	NA
Majumder et al. [27]	Mesenteric; inguinal	Enzymatic	DNase I; Liberase; Dispase	RPMI	CD45; CD31; PDPN;	Microbeads isolation	>98
Masters et al. [28]	Mesenteric; popliteal	Enzymatic	Liberase TL; Benzonuclease	RPMI-1640	CD45; CD31; PDPN	Microbeads isolation	>90
Schaeuble et al. [29]	Peripheral (axillary, brachial, inguinal)	Enzymatic	Collagenase IV; DNase I	DMEM (2% FCS)	CD45; CD31; PDPN	Microbeads isolation	≥90
Dubrot et al. [30]	Skin-draining	Enzymatic	Collagenase D; DNase I	HBSS	CD45; CD31; PDPN	Cell sorting	NR
Knoblich et al. [31]	NR	Enzymatic	Collagenase P; DNase I; Dispase	α-MEM (10% FBS)	CD45; PDPN	NR	99
Maaraouf et al. [32]	Kidney	Enzymatic	Collagenase P; DNase I; Dispase II	DMEM (10% FBS)	CD45; CD31; PDPN	NR	NR
Chung et al. [33]	Peripheral (cervical, axial, brachial, inguinal)	Enzymatic	Collagenase IV; DNase I	DMEM (2% FBS)	CD45; CD31; PDPN	Cell sorting	NR
Gao et al. [34]	Inguinal	Enzymatic	Collagenase IV; DNase I	RPMI-1640 (2% FBS)	CD45; CD31; PDPN	NA	NA
Pazstoi et al. [35]	Mesenteric	Enzymatic	Collagenase P; Dispase; DNase I	RPMI-1640	CD45; CD31; PDPN	Cell sorting	91–97
Valencia et al. [36]	Mesenteric	Mechanical disruption	NR	RPMI-1640	CD45, CD31, PDPN	NR	NR
Yu, M. et al. [37]	Axillary; brachial; inguinal	Enzymatic	Collagenase P; Dispase; DNase I	DMEM (10% FBS)	CD45; CD31; PDPN	Cell sorting	>95
Gil-Cruz et al. [38]	Mesenteric	Enzymatic	Collagenase D; DNase I	RPMI-1640 (2% FCS)	CD45; CD31; PDPN	Cell sorting	NR
Novkovic et al. [39]	Inguinal	Enzymatic	Collagenase P; DNase I	RPMI (2% FCS)	PDPN	NA	NA
Royer et al. [40]	Mandibular	Mechanical disruption	NR	RPMI-1640 (10% FBS)	NR	NR	NR
Takeda et al. [41]	Mesenteric; peripheral; brachial	Enzymatic	Collagenase P; Dispase; DNase I	RPMI-1640	CD45; CD31; PDPN	Cell sorting	NR

Abbreviations—Ref.: reference; NR: not reported; NA: not applicable; FBS: fetal bovine serum; FCS: fetal calf serum; PDPN or gp38: podoplanin; NC: not cultivated.

**Table 3 cells-10-01150-t003:** Characteristics of the main type of immune cells used for analysis with fibroblastic reticular cells.

Ref.	Source of Cells	Cell Type	Separation Technique	Immune Cell Preservation Solution and Supplementation	Immune Cell Immunophenotypic Characterization
Aparicio-Domingo et al. [18]	LN	CD8+ T cells	Non selection performed	DMEM (2% FCS)	CD45, CD8α, CD4, TRCαβ
Dertschnig et al. [19]	LN	T cells	CD3 negative selection followed by CD4 and CD8α positive selection (MicroBeads—Myltenyi)	NR	CD45, CD45.1, CD3, CD4, CD8α, CD62L, CD44, CD69, CD127, Vα2, Vβ5
Eom et al. [20]	LN	NA	NA	NA	CD45, CD3, CD8
Gonzalez et al. [21]	Spleen	CD8+ T cells	CD8 isolation by negative selection (Microbeads—MojoSort)	NR	CD45, CD8, CD44, CD25
Knop et al. [22]	LN; spleen	T cells and NK	CD8α positive selection (MicroBeads—Myltenyi)	RPMI	CD45, CD3, CD4, CD5, CD8α, CD62L, Bcl-2, CD127, Nk1.1, RORγt
Perez-Shibayama et al. [23]	LN; spleen	T cell subsets and exhaustion	No selection performed	RPMI	CD45.1, CD45.2, CD45R, CD8α, CD8β, CD3e, CD44 CD62L, PD-1, PDL1
Brown et al. [24]	LN	CD8+ T cells	CD8α positive selection (MicroBeads—Myltenyi)	RPMI; α-MEM	CD45.1, CD45.2, CD3, CD4, CD8, CD275, CD28, CD44
Kasinath et al. [25]	LN; spleen	CD4+ T cells	No selection performed	NR	CD45, CD3, CD4, CD44, CD62L, IL-17A
Kelch et al. [26]	LN	NA	NA	NA	NA
Majumder et al. [27]	LN	T and B cells	NR	NR	CD45, CD45.2, CD4, B220, IL-17A, IL-17R
Masters et al. [28]	LN; peripheral blood	CD8+ T	CD8 isolation by negative selection (Microbeads—MojoSort)	NR	CD45, CD45.1, CD45.2, CD69, CD8α
Schaeuble et al. [29]	LN; spleen	T cells	No selection performed	RPMI	CD45, CD3, CD4, CD8α, CD44, CD62L, CD279, FoxP3, CD25
Dubrot et al. [30]	LN; spleen	T cells, B cells, Treg, and DC	Pan T isolation by negative selection (MicroBeads—Myltenyi)	NR	CD45, CD44, CD3, CD4, CD8α, FOXp3, Ly5.1, CD11b, CD19, CD25, CD62L, PDCA-1, PD-1, IL-17, IFNγR
Knoblich et al. [31]	LN; tonsils	T cells	Pan T isolation by negative selection (MicroBeads—Myltenyi)	NR	CD45, CD3, CD4, CD8, CD62L, CD27, CD45RO, CD25
Maaraouf et al. [32]	Spleen	T cells	Pan T isolation by negative selection (MicroBeads—Myltenyi)	NR	CD45, CD4
Chung et al. [33]	Spleen; peripheral blood	T cells, B cells, FDCs, Treg, and DCs	T cell Thy.1 selection (Microbeads—StemCells Technologies)	NA	CD45, CD3, CD4, CD8, FOXp3, CD157, CD19, B220, CD44, CD62L, CD11c, CD11b, CD169, CD21/35, F4/80, TCRβ
Gao et al. [34]	LN	T cells	NR	NR	CD45, CD4, CD8
Pazstoi et al. [35]	LN	T cells	CD4 positive selection (Microbeads—Myltenyi)	EX VIVO	CD45, CD45.2, CD4, CD2, CD9, CD24, CD25, CD63
Valencia et al. [36]	LN	CD4+ T cells	CD4 naïve T cell negative selection (Microbeads—Myltenyi)	RPMI (10% FCS)	CD45, CD44, CD4
Yu, M. et al. [37]	LN	T cells	Pan T cell negative selection (Microbeads—StemCells Technologies)	RPMI (10% FBS)	CD45, CD45.1, CD45.2, CD3, CD4, CD8α, CD25, CD69, CD44
Gil-Cruz et al. [38]	PP; LN	T cells, B cells, NK cells, Treg, and ILCs	NR	RPMI (10% FCS)	CD45, CD3e, CD4, CD8α, EOMES, FoxP3, B220, CD19, CD127, CD62L, CD44, CD69, F4/80, IL-17A, IL-7Rα, GATA3, RORγt, IL-15RαIL-15Rβ, NKp46, NK1.1
Novkovic et al. [39]	LN; Spleen	DCs and T cells	NR	RPMI (2% FCS)	CD45, CD3, CD8, CD4, CD11c, MHCII
Royer et al. [40]	SLOs	CD8+ T cells	CD8 positive selection (Microbeads—Myltenyi)	RPMI (10% FBS)	CD45, CD3, CD4, CD8
Takeda et al. [41]	LN; Spleen	T cells, B cells	CD4 naïve T cell negative selection (Microbeads—Myltenyi)	RPMI	CD4, CD8, B220, CD44

Abbreviations:—Ref.: reference; NR: not reported; NA: not applicable; FCS: fetal calf serum; PP: Peyer’s Patches; LN: lymph nodes; SLOs: secondary lymphoid organs; DCs: dendritic cells; NK: natural killer cells.

**Table 4 cells-10-01150-t004:** Main characteristics of the studies used to assess the influence of fibroblastic reticular cells on the activation, expansion, or suppression of immune responses.

Ref.	Trial Types	Study Target	Time of Intervention	Main Performed Evaluations	Results	FRC Role in Immune Response
Aparicio-Domingo et al. [18]	IL-33-GFP reporter mice	LCMV	3 days/w for 2 weeks	FC and RNA sequencing	FRC is one important IL-33 source in LNs, vital for driving acute and chronic antiviral T cell responses.	Anti-viral response
Dertschnig et al. [19]	FRC and DC ablation in vivo; identification of PTA regulatory genes; BTM model induction	GvHD	2 weeks	FC, RNA sequencing, confocal microscopy	The loss of PTA presentation by FRCs during GVHD leads to permanent damage in their networks in lymphoid tissues.	Control of peripheral tolerance
Eom et al. [20]	Identification of distinctive subpopulations of CD90+ SCs present in melanoma-infiltrated LNs	Melanoma	NA	FC, gene expression	There are several distinct subsets of FRCs present in melanoma-infiltrated LNs. These FRCs may be related to cancer metastasis invasion and progression by avoiding T cells through secreted factors.	Lymph node invasion metastasis and its correlation with FRC gene expression.
Gonzalez et al. [21]	Tissue-engineered stromal reticula and FRC/T cell co-culture	Type 1 diabetes	NA	FC, immunofluorescence, imaging	FRCs modulate their interactions with autoreactive T cells by remodeling their reticular network in LNs. FRC with decreased contractility through gp38 downregulation, can loosen/relax their network, potentially decreasing FRC tolerogenic interactions with autoreactive T cells and promoting their escape from peripheral regulation in LNs.	Role of FRCs on tolerance and T1D
Knop et al. [22]	IL-7^fl/fl^ mice and adoptive T cell transfer	NA	NA	FC	IL7, produced by LN FRCs-regulated T cell homeostasis, is crucial for T_CM_ maintenance.	IL7 produced by LN FRCs is crucial for T_CM_ maintenance
Perez-Shibayama et al. [23]	LCMV-infected mice, FRC ex vivo restimulation and cytokine production	LCMV Armstrong	8 d	FC	IFNAR-dependent shift of FRC subsets toward an immunoregulatory state reduces exhaustive CD8+ T cell activation.	IFN type 1 influences FRC peripheral tolerance
Brown et al. [24]	FRC/T cell co-cultures	Influenza and LCMV infection	NR	FC and RNA sequencing	FRCs play a role over restricting T cell expansion—they can also outline the fate and function of CD8+ T cells through their IL-6 production.	FRCs influence the CD8 T cells fate
Kasinath et al. [25]	Mouse FRC depletion and treatment with anti-PDPN antibody	Crescentic Glomerulonephritis (GN)	3 d	FC and gene expression	Removal of kidney-draining lymph nodes, depletion of fibroblastic reticular cells, and treatment with anti-podoplanin antibodies each resulted in the reduction of kidney injury in GN.	Role of FRCs and PDPN expression in GN
Kelch et al. [26]	3D imaging and topological mapping	NA	NA	EVIS imaging and confocal microscopy	T cell zones showed homogeneous branching, conduit density was significantly higher in the superficial T cell zone compared with the deep zone. Although the biological significance of this structural segregation is still unclear, independent reports have pointed to an asymmetry in cell positioning in both zones. Naive T cells tend to occupy the deep TCZ, whereas memory T cells preferentially locate to the superficial zones, and innate effector cells can often be found in the interfollicular regions.	FRC conduits and their distribution inside LNs
Majumder et al. [27]	Metabolic assay	Experimental autoimmune encephalomyelitis	7 d	FC, immunoblotting, siRNA transfection	During Th17 differentiation in LNs, IL-17 signals to FRCs and impacts LN stromal organization by promoting FRC activation through a switch on their phenotype from quiescence to highly metabolic.	FRCs are impacted by metabolic alterations driven by IL-17
Masters et al. [28]	FRC-mediated T cell proliferation inhibition and T cell survival assays	Aging and influenza infection	NR	FC	Age-related changes in LN stromal cells may have the largest impact on the initiation of the immune response to influenza infection, and may be a factor contributing to delayed T cell responses to this virus.	Aging impacts the adaptive anti-viral immune response initiation in LN
Schaeuble et al. [29]	*Nos2^−/−^*, *COX2^−/−^* mice and FRC/T cell co-culture	COX/Prostaglandin E2 pathway	4 d	FC	FRCs constitutively express high levels of COX2 and its product PGE_2_, thereby identified as a mechanism of T cell proliferation control.	PGE_2_ and COX2 pathways in FRCs are implicated in the control of T cell proliferation
Dubrot et al. [30]	Adoptive transfer T cells in RAG^−/−^ mice and Treg suppression assay	MHC II-induced expression by FRC and LEC and its impact on autoimmunity	5 d	FC	LNSCs inhibit autoreactive T-cell responses by directly presenting antigens through endogenous MHCII molecules.	Control of peripheral tolerance in autoimmunity
Knoblich et al. [31]	T cell and CAR T cell activation assay	COX/Prostaglandin E2, iNOS, IDO and TGF-β pathways in FRCs,	NA	FC and RNA sequencing	FRCs block proliferation and modulate differentiation of newly activated naïve human T cells, without requiring T cell feedback.	FRCs used several pathways to control T cell proliferation
Maaraouf et al. [32]	FRC labeling and injection into mice	Ischemic-reperfusion injury (IRI)	NR	FC, electron and confocal microscopy	Depletion of FRCs reduced T cell activation in the kidney LNs and ameliorated renal injury in acute IRI.	Role of FRCs in IRI
Chung et al. [33]	FRC/T cell co-culture	GvHD	4 h and 3 h	FC	FRCs delivered NOTCH signals to donor alloreactive T cells at early stages after allo-BMT to program the pathogenicity of these T cells.	Role of FRC NOTCH-signaling in activating alloreactive T cells
Gao et al. [34]	FRC expression and secretion of Interleukin 7	Tumor-draining LNs	NA	FC	LN tumor-infiltrating cells decreased the FRC population and IL-7 secretion, leading to declined numbers of T cells in TDLNs. This may partly explain the weakened ability of immune surveillance in TDLNs.	Role of IL-7 secretion by FRCs and its impact on tumor-draining LNs
Pazstoi et al. [35]	Treg induction in presence of FRC microvesicles.	FRC microvesicles (MVEs)	NA	FC and RNA sequencing	Stromal cells originating from LNs contributed to peripheral tolerance by fostering de novo Treg induction by MVEs carrying high levels of TGF-β.	Role of FRC MVEs in inducing peripheral tolerance
Valencia et al. [36]	FRC/T cell co-culture	COX 2/Prostaglandin E2, iNOS, IDO and TGF-β pathways in FRCs	6 h	FC	COX2 expression was detected in human FRCs but was not considerably upregulated after inflammatory stimulation, concluding that human and murine FRCs would regulate T lymphocytes responses using different mechanisms.	Role of FRCs integrating innate and adaptive immune responses and balancing tolerance and immunogenicity
Yu, M. et al. [37]	FRC/T cell co-culture	COX 2/Prostaglandin E2 pathway in FRCs	NA	FC, WB	Hyperactivity of COX-2/PGE_2_ pathways in FRCs is a mechanism that maintains peripheral T cell tolerance during homeostasis.	PGE_2_ and COX2 pathways in FRCs are implicated in the control of T cell proliferation.
Gil-Cruz et al. [38]	ILC1 and NK cells regulation	FRC secretion of IL-15	3 h	FC	FRC secretion of IL-15 regulates homeostatic ILC1 and NK cell maintenance.	Role of FRCs in innate in immunity
Novkovic et al. [39]	FRC network topological analysis	FRC network	NA	Intravital TPM with morphometric 3D reconstitution analysis.	Physical scaffold of LNs formed by the FRC network is critical for the maintenance of LN functionality.	FRC network disruption impacts the immune response
Royer et al. [40]	Adoptive transfer of T cells and T cell response to herpesvirus-associated lymphadenitis	HSV-1	4 h	FC	Dissemination of the virus to secondary lymphoid organs impairs HSV-specific CD8+ T cell responses by driving pathological alterations to the FRCs conduit system, resulting in fewer HSV-specific CD8+ T cells in circulation.	Role of FRC in virus-specific T CD8 response
Takeda et al. [41]	Lymphocyte migration	Ablation of LPA-producing ectoenzyme autotaxin in FRCs	NA	FC, IMS, Intravital TPM	LPA produced by LN FRCs acts locally to LPA2 to induce T cell motility.	Role of FRCs in T cell local migration

Abbreviations—Ref.: reference; NR: not reported; NA: not applicable; FC: flow cytometry; WB: Western blotting; IMS: imaging mass spectrometry; Intravital TPM: intravital two-photon microscopy.

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
