# Peer review of "LN-Derived Fibroblastic Reticular Cells and Their Impact on T Cell Response—A Systematic Review"

_cells, 2021, doi:10.3390/cells10051150_

Round 1
Reviewer 1 Report
Brief summary:
The review by Ferreira et al., overviews recent studies (2016-2020) on fibroblastic reticular cells (FRC) of the T cell zone in murine and human lymh nodes and other secondary lymphoid organs. The review focus on the influence of FRC on T cell immune responses in homeostatic and disease settings.
Broad comments:
This review comprehensively cites the relevant literature in the field, which comprises several new and exciting findings on the function of stromal lymph node cells. It would be helpful if the different findings of the reviewed studies were more integrated and commented by the authors with their strengths and limitations/weaknesses . The reasoning for the selected parameters that were analyzed in the results section and the meaning of the outcome of theses analyses is not completely clear. The concrete new overall findings through this review by the combined literature research should be more clearly carved out and unknowns named specifically. Regarding the PRISMA 2009 checkklist, not all points were adressed (eg, #1 Title: „Identify the report as a systematic review, meta-anaalysis, or both“). A combination of grammar mistakes (eg, confused singular/plural, pronouns, tenses), very long sentences (eg, lines 378 – 387, 476-483, 604-612), leftovers from the editors template (lines 141-142) and a blank page made the text a bit difficult to read.
Specific comments:
Abstract:
- What’s the reasoning behind chosing years 2016 – 2020? Please also state if/how you consider/integrate older but major breakthrough studies?
- Please state the total number of studies analyzed.
Introduction:
- Please mention tertiary lymphoid organs or comment why they were not considered
- Line 46: the references stated do not address the deletion of self-reactive T cells
- Line 51: I do not get the meaning of this sentence, please rephrase
- Line 59: „lymphocyte fate“ – since the review only focuses on T cells, please write „T cells“
- Line 60: „how they could be maipulate“ – I found no hints about this point later in the text?
Material and Methods:
- Section 2.2 Inclusion criteria: would be helpful to have a comment on how you dealt with results from in vivo studies or studies that combined both in vitro and in vivo experiments (were they covered by "network analysis"?).
Results:
- Figure 1: first line – why is the total 75 when 32 + 43 + 100 equals 175?
- Section 3.2 Characterisitics of the animal or human cell models and Section 3.2 General immunophenotypic characteristics of fibroblastic reticular cells:
Given the complexities of the studies, that use different cominations of cell models and of many of the other analyzed parameters, the benefit of listing these indivdiuals parameters seperately becomes not immediatley clear to me. Which conclusions can be drawn? What’s the benefit of the text over Tables 1 and 2? Why did you chose to analyze the reported parameters (e.g. use of RPMI or DMEM; Novkovic et al. using only PDPN as FRC marker?).
- Section 3.4 lines 305 – 327 seem to state the same content as Table 3 and if so, the text can be removed
- Section 3.4.2 Lines 363-364: „after ageing-related changes, FRC have an impact on the ..immunresponse to influenza“: I understand, that FRC have always an impact, just reduced function with age. This should be rephrased.
- Lines 370: „…in accordance with Knoblich et al. which also claimes that FRCs block proliferation and modulate differentiation…“: I find this statement confusing since different mechanisms were reported for human and murine FRC (althouth with similar final outcome). Please rephrase.
Discussion:
- Fig. 2: cplease learly state whether only newly discovered functions are depicted from the 2016-2020 period literature review? Maybe add another box with homeostetic functions so that changes in disease become more obvious? Differentiate between murine/human mechanisms or indicate were results are derived from mice if relevant
- I missed a comment on how different FRC subsets and their location may play a role for the stated findings.
- Line 472: please specifically name common and conflicting results. A judgement on the importance of individual studies and their limitations would be required.
- Lines 501 - 506: Please state explicitely, which Masters et al. findings are supported by Chai et al.
Author Response
Cells-1154712
LN-derived fibroblastic reticular cells and their impact in T cells response – A systematic review
Reviewer #1
- Broad comments:
This review comprehensively cites the relevant literature in the field, which comprises several new and exciting findings on the function of stromal lymph node cells. It would be helpful if the different findings of the reviewed studies were more integrated and commented by the authors with their strengths and limitations/weaknesses. The reasoning for the selected parameters that were analyzed in the results section and the meaning of the outcome of theses analyses is not completely clear. The concrete new overall findings through this review by the combined literature research should be more clearly carved out and unknowns named specifically. Regarding the PRISMA 2009 checkklist, not all points were adressed (eg, #1 Title: „Identify the report as a systematic review, meta-anaalysis, or both“). A combination of grammar mistakes (eg, confused singular/plural, pronouns, tenses), very long sentences (eg, lines 378 – 387, 476-483, 604-612), leftovers from the editors template (lines 141-142) and a blank page made the text a bit difficult to read.
Answer: Thank you for your suggestions. We have included a systematic review in the article title in order to accomplish with PRISMA guidelines. We also have revised the English language in the manuscript in order to make it more adequate
Specific comments:
- Abstract:
- What’s the reasoning behind chosing years 2016 – 2020?
- Please also state if/how you consider/integrate older but major breakthrough studies?
- Please state the total number of studies analyzed.
Answer: We have included the requested information in the abstract:
“We searched articles indexed and recently published, including years between 2016 and 2020 in PubMed, Scopus and Cochrane following the PRISMA guidelines. We have found 175 articles published in literature, but only 24 articles fulfill our inclusion criteria and were discussed here. Other articles important in the built knowledge of FRCs were included in the introduction and discussion.”
- Introduction:
Please mention tertiary lymphoid organs or comment why they were not considered
Answer: Regarding tertiary lymphoid organs they were not included because there is no report about the presence of FRCs in these organs.
Line 46: the references stated do not address the deletion of self-reactive T cells
Answer: In line 46, we have included another 03 references “Fletcher AL, Malhotra D, Turley SJ. Lymph node stroma broaden the peripheral tolerance paradigm. Trends Immunol. 2011 Jan;32(1):12-8. doi: 10.1016/j.it.2010.11.002. and “Gardner JM, Devoss JJ, Friedman RS, Wong DJ, Tan YX, Zhou X, Johannes KP, Su MA, Chang HY, Krummel MF, Anderson MS. Deletional tolerance mediated by extrathymic Aire-expressing cells. Science. 2008 Aug 8;321(5890):843-7. doi: 10.1126/science.1159407. “ and Nadafi, R.; Gago de Graça, C.; Keuning, E.D.; Koning, J.J.; de Kivit, S.; Konijn, T.; Henri, S.; Borst, J.; Reijmers, R.M.; van Baarsen, L.G.M., et al. Lymph Node Stromal Cells Generate Antigen-Specific Regulatory T Cells and Control Autoreactive T and B Cell Responses. Cell Rep 2020, 30, 4110-4123.e4114, doi:10.1016/j.celrep.2020.03.007.
Line 51: I do not get the meaning of this sentence, please rephrase
Answer: In line 51 we have rephrased the sentence “The articles reviewed here used several animal models of disease and/or genetically modified as a tool to investigate FRCs effect on T cell. These articles also approach and clarify mechanisms involved in T cells proliferation or differentiation in subsets with regulatory, effector or memory profile.
Line 59: „lymphocyte fate“ – since the review only focuses on T cells, please write „T cells“
Answer: In line 59 - lymphocytes fate was replaced by T cells fate.
Line 60: „how they could be maipulate“ – I found no hints about this point later in the text?
Answer: In line 60 - how they could be manipulate, we are referring to articles that suggested treatments in order to ameliorate certain conditions, as one article that used anti-podoplanin antibodies to treat the model of renal injury.
- Material and Methods:
- Section 2.2 Inclusion criteria: would be helpful to have a comment on how you dealt with results from in vivo studies or studies that combined both in vitro and in vivo experiments (were they covered by "network analysis"?).
Answer: We accepted all studies with in vitro or/and in vivo covered by the network analysis. And we have modified this sentence in the article.
- Results:
- Figure 1: first line – why is the total 75 when 32 + 43 + 100 equals 175?
Answer: Thank you for this observation, you are right; the correct number of articles is 175 and this number was corrected in the figure 1.
- Section 3.2 Characterisitics of the animal or human cell models and Section 3.2 General immunophenotypic characteristics of fibroblastic reticular cells:
Given the complexities of the studies, that use different combinations of cell models and of many of the other analyzed parameters, the benefit of listing these individuals parameters seperately becomes not immediatley clear to me. Which conclusions can be drawn? What’s the benefit of the text over Tables 1 and 2? Why did you chose to analyze the reported parameters (e.g. use of RPMI or DMEM; Novkovic et al. using only PDPN as FRC marker?).
Answer: There are difficulties associated with characterization and isolation of these cells. So, we imagined that summarizing technical details in how authors isolated, cultivated and characterized these cells this would be helpful for other investigators. Even though, media preferences looks to be a simple detail, when working with co-cultures (FRCs and T cells), is not so obvious which media to choose.
In addition, it is also important to show that there are variation between protocols, what would maybe responsible for different or conflicting results described in literature.
Regarding the use of only PDPN to characterize FRCs by NovKovic et al, it is justified by the type of study that they performed, that was intravital microscopy, so I removed this comment from the text in order to not lead to a misunderstood about it.
- Section 3.4 lines 305 – 327 seem to state the same content as Table 3 and if so, the text can be removed
Answer: Yes the content was very similar to the content of the Table 3, so this part was removed from the results section.
- Section 3.4.2 Lines 363-364: „after ageing-related changes, FRC have an impact on the .immunresponse to influenza“: I understand that FRC have always an impact, just reduced function with age. This should be rephrased.
Answer: We have altered this phrase to “Masters et al., reported that after aging-related changes, FRCs have their impact altered on the initiation of the immune response to influenza infection, and this contribute to delayed T lymphocytes responses to this vírus”
- Lines 370: „…in accordance with Knoblich et al. which also claimes that FRCs block proliferation and modulate differentiation…“: I find this statement confusing since different mechanisms were reported for human and murine FRC (althouth with similar final outcome). Please rephrase.
Answer: I have modified the phrase “Knoblich et al. also demonstrated that FRCs controls T cells proliferation and modulate their differentiation”…….
- Discussion:
- 2: please clearly state whether only newly discovered functions are depicted from the 2016-2020 period literature review? Maybe add another box with homeostetic functions so that changes in disease become more obvious? Differentiate between murine/human mechanisms or indicate were results are derived from mice if relevant
Answer: Not only newly discoveries were FRCs functions were depicted, but only original articles were included from the period of 2016 and 2020. We have covered the last 5 years of original publications about FRCs and T cells interactions. We have added a box (2F) about the FRCs role in homeostasis. In Figure 2B we have showed the differences between murine and human mechanisms.
- I missed a comment on how different FRC subsets and their location may play a role for the stated findings.
Answer: I have included an observation in the discussion section “FRC is a specific subset of stromal cells present in the lymph node, and they are precisely located in the T cell zone. There are other stromal cells subsets described in lymph node, as double negative cells, follicular dendritic cells, blood endothelial cells, lymphatic endothelial cells and others that are not discussed in this review”
- Line 472: please specifically name common and conflicting results. A judgement on the importance of individual studies and their limitations would be required.
Answer: We have modified the comment on the discussion section “All these variations between the models studied, cells origin and characterization, sometimes lead to different conclusions turning the comparison between studies difficult or conflicting, as the role of FRCs in T cells proliferation, sometimes described as stimulators and other times as limiting.”
- Lines 501 - 506: Please state explicitely, which Masters et al. findings are supported by Chai et al.
Answer: On the Chai et al article they highlight the importance of FRCs maturation process and the production of homeostatic chemokines for the anti-viral, in the Masters article even though in a different context, they also report the importance of these same homeostatic chemokines for the success of the anti-viral response. I also have included an observation in the discussion section of the manuscript.
Reviewer 2 Report
In this systematic review Ferreira et al examine the role of fibroblastic reticular cells and their impact in immune cell functions. The review is interesting and timely, the authors discuss in depth the new emerging role of structural cells, and in particular of FRC, in the regulation of immune response.
Before acceptance I would suggest some changes:
Since the authors focused on T cells the title should be changed in “LN-derived fibroblastic reticular cells and their impact on T cell responses”
Several important and recent references are missing:
- Krausgruber et al, Structural cells are key regulators of organ-specific immune responses. Nature 2020
- Akshay T. Krishnamurty. Lymph node stromal cells: cartographers of the immune system. Nat Immunol
- Denton, A. E. et al. Embryonic FAP+ lymphoid tissue organizer cells generate the reticular network of adult lymph nodes. J. Exp. Med. 216, 2242–2252 (2019)
- Martinez V et al. Fibroblastic Reticular Cells Control Conduit Matrix Deposition during Lymph Node Expansion. 2019
- Nadafi et al, Lymph Node Stromal Cells Generate Antigen Specific Regulatory T Cells and Control Autoreactive T and B Cell Responses
It would be helpful, to make the text more structured by subdividing certain sections, particularly the sections on “The role of FRCs in immune response (paragraph 3.4.2)”. I would structure in 1) the role of FRCs in infection, 2) in inflammation/autoimmunity, 3) in cancer. This could facilitate the reading.
Abstract: please correct “lymph” line 16; what does abbreviation LPA mean? (line 28). A list of abbreviation could be useful.
Introduction: add reference line 37
Figure: Resolution is poor and difficult to read. The font size is too small.
English needs to improve, there are several grammatical mistakes throughout the text. E.g. line 47 page 2 “needs” is without “s”.
Author Response
In this systematic review Ferreira et al examine the role of fibroblastic reticular cells and their impact in immune cell functions. The review is interesting and timely, the authors discuss in depth the new emerging role of structural cells, and in particular of FRC, in the regulation of immune response.
Before acceptance I would suggest some changes:
- Since the authors focused on T cells the title should be changed in “LN-derived fibroblastic reticular cells and their impact on T cell responses”
Answer: Thank you for the suggestion, the title was modified to” LN-derived fibroblastic reticular cells and their impact in T cells response – A systematic review”
- Several important and recent references are missing:
- Krausgruber et al, Structural cells are key regulators of organ-specific immune responses. Nature 2020
- Akshay T. Krishnamurty. Lymph node stromal cells: cartographers of the immune system. Nat Immunol
- Denton, A. E. et al. Embryonic FAP+ lymphoid tissue organizer cells generate the reticular network of adult lymph nodes. J. Exp. Med. 216, 2242–2252 (2019)
- Martinez V et al. Fibroblastic Reticular Cells Control Conduit Matrix Deposition during Lymph Node Expansion. 2019
- Nadafi et al, Lymph Node Stromal Cells Generate Antigen Specific Regulatory T Cells and Control Autoreactive T and B Cell Responses
Answer: Answer: Thank you for the suggestion. Even though these articles were not related with FRCs and T cells interactions and were not displayed by our searching strategy, we have considered all of them very important for the knowledge about FRCs and included them in the introduction and/or discussion.
- It would be helpful, to make the text more structured by subdividing certain sections, particularly the sections on “The role of FRCs in immune response (paragraph 3.4.2)”. I would structure in 1) the role of FRCs in infection, 2) in inflammation/autoimmunity, 3) in cancer. This could facilitate the reading.
Answer: We have structured the suggested section in topics.
- Abstract: please correct “lymph” line 16; what does abbreviation LPA mean? (line 28). A list of abbreviation could be useful.
Answer: We have corrected the meaning for lymph and included the meaning of LPA (Lysophosphatidic acid) in the abstract.
- Introduction: add reference line 37
Answer: A reference was added.
- Figure: Resolution is poor and difficult to read. The font size is too small.
Answer: Thank you for this observation. We have increased the figure resolution and also the font size.
- English needs to improve, there are several grammatical mistakes throughout the text. E.g. line 47 page 2 “needs” is without “s”.
Answer: Thank you for pointing out these inaccuracies. We have revised the English language in the manuscript in order to make it more adequate
Reviewer 3 Report
In this paper, the authors made a huge work by performing a systematic review of the methods employed to investigate the role of fibroblastic reticular cells to control the immune response. Nevertheless, particularly in the “results” section, it appears as a heavy list rather hard to follow for a not-expert reader and should be lightened, because most data are already included in tables. The discussion is interesting by describing different but also contradictory roles of FRC in several biological contexts. However, due to the wide amount of data arising from this review, I suggest a brief summary of most relevant functions of FRC, as illustrated in Figure 2.
Author Response
- In this paper, the authors made a huge work by performing a systematic review of the methods employed to investigate the role of fibroblastic reticular cells to control the immune response. Nevertheless, particularly in the “results” section, it appears as a heavy list rather hard to follow for a not-expert reader and should be lightened, because most data are already included in tables. The discussion is interesting by describing different but also contradictory roles of FRC in several biological contexts. However, due to the wide amount of data arising from this review, I suggest a brief summary of most relevant functions of FRC, as illustrated in Figure 2
Answer: Thank you for these observations. We have modified the results section, excluded a very repeated parts and itemized part of the section to make it more clear and easier to read. At the end of discussion we have summarized the most important finds.
Round 2
Reviewer 1 Report
Broad comments
In the revised version of the manuscript, some details improved, mainly those linked to the specific comments from the first review (e.g., additional references were included, a wrong number in figure 1 was corrected, etc).
However, I find it difficult to identify the main result of this systematic review. The abstract lists the percentage of FRCs studies that use a certain “strategy” for analysis. It gives results from 15 studies but misses 9 studies (out of 24 in total). For me, I don’t see how the fact that, for instance, 21% of studies investigate FRC function in viral infections substantially adds to the understanding of FCR biology.
The title suggest that the main result/objective of this systematic review is the FRCs’ impact on T cells. However, the abstracts does not mention how FRCs affect T cells.
In my view, the experimental details listed in Tables 1 to 4 are not sufficiently linked to the respective findings of the individual studies. Of course, experimental details are tremendously important, but usually function in an experimental system as a whole, e.g. a certain digestive enzyme may destroy an epitope for antibody staining etc. The authors miss to mention the extra/additional/new value from these tables, e.g. how they provide new insights in this review on FRC function or how researchers could use these to improve future studies or what current limitations are.
In the sections that discuss the findings of the individual studies (e.g. starting with results part 3.5), I find that important aspects are missing when I try to understand individual sentences in the text. I usually need to read the original literature to understand the author’s statements and put it into context. Examples are given in the specific comments.
I do not find that the language improved. I still find it difficult to follow the text given mixed, inconsistent use of tenses. Many sentences are incomplete and/or very long (6 or more lines, e.g. lines 338 – 343). Many sentences have grammar errors and even headings have spelling errors (e.g. 3.3.2- Tissue disagregation type). There is a leftover from the journals document template at the end of the results section 3.1 (lines 156-158). Writing styles vary a lot (e.g. T-cell vs. T cell, inconsistent use of the abbreviation FRCs, single numbers with or without “0” in front, e.g. 04 studies vs 3 studies, incomplete author names like Aparicio-Domingo, etc.). The authors repeat whole sentences identically in result and discussion part. Although these aspects may not be part of the scientific review, combined they do not illustrate great care by the authors.
Overall, I do not recommend publication in the present form.
Specific comments:
Introduction
In my opinion, a clear statement on the rationale of the review, its objective and/or answered research questions is missing (similar as stated in broad comments for title/abstract).
Materials and Methods
Section: 2.1: SCOPUS: the search term “T lymphocyte” occurs twice (plural missing?)
Results
Section 3.2: Line 168 “The host used in the majority of the studies reviewed here were mice C57BL/6 (79%)”. Lines 173 – 175 “Besides that, 4 studies (19%) used the background C57BL/6J [18,25,26,41], 3 (13%) used C57BL/6N [22,38,39], 2 used 174 BALB/c2 [33,35], and C57BL/6N-Tg [38,39] (10% each).”
I find it difficult to combine these statements.
Section 3.5.1:
Lines 344 – 346 contains a plagiat from the abstract of the cited study by Brown et al., 2019 (Ref. 24): “…that FRCs play a role beyond a regulatory 344 role restricting T cell expansion—they also shape the fate and function of CD8 T lymphocytes…”
Lines 246 – 348 “Masters et al., reported that after aging-related changes, FRCs have their impact altered on the initiation of the immune response to influenza infection, and this contribute to delayed T lymphocytes responses to this virus [28].” From my understanding, the study by Masters et al. only concludes, that aged FRCs show reduced proliferation and may play a role in reduced immune responses. The authors state in their abstract, that “aged FRCs did not appear to be a contributing factor in the reduced proliferation of young T cells transferred into aged LNs after influenza infection”.
Section 3.5.2:
In Fig. 2B, influences on CD8+ T cells are listed but Knoblich et al., 2018 (Ref. 31) also shows inhibition of proliferation of CD4+ T cells. PEG2 control appears to be most important in homeostasis (Schaeuble et al., 2019, Ref. 29), but is only listed in Fig. 2B and in the text in the subsection about “inflammation” which I find misleading. The study by Yu et al., 2017 does not study FRC characteristics in inflammation at all.
Line 360 – 361 “Valencia et al., discuss the differences between mice and humans regarding COX inflammatory pathway...” Actually, the Cox pathway seems to be very similar in mice and humans while humans lack NOS2 induction, which Is important in mice. I do not get this information from the text in the manuscript.
Section 3.5.1.7
Lines 400 – 401: “Dubrot et al. showed a mechanism of T lymphocytes proliferation inhibition by the induced expression of MHC II”. This sentence alone is not understandable to me. Upon checking the cited literature, I see it’s about Tregs and that it is actually not clear whether the observed effect is due to FRC or other lymph node stroma cells. I would expect a review to put such findings into context.
Lines 401 – 403: “Pazstoi et al. described FRCs contribute to peripheral tolerance by fostering de novo Treg induction by MVEs carrying high levels of TGF-β.” The abbreviation MVE is not explained and not present in the cited study. The cited study shows that only mesenteric lymph nodes seem to produce microvesicles with TGFb in mice while peripheral lymphnodes do not have this function. I find such details important to be able to better judge the current state of knowledge.
Lines 403 - 404: “Gil-Cruz et al. comment the role of FRCs on innate lymphocytes ILC1 and NK through IL-15 secretion [38]”. In this study, IL-15 knock-out also seems to result in decreased Tregs and increased inflammatory TH1 cells. Since the review is about T cells, I would expect that this finding is discussed and not ILC1?
Discussion
Lines 439 – 450 could be removed.
Lines 451 – 454 “Aparicio et al. in a LCMV study concluded that FRCs displayed a stimulatory role, being an important source of IL-33 in the lymph node and vital for driving acute and chronic antiviral T-cell responses.” This sentence is identical to lines 338 – 340.
Lines 468 – 469 “Brown et al. [24] showed that FRCs play a role beyond a regulatory role restricting T cell expansion” are identical to lines 344-345.
Reviewer 2 Report
The authors have answered all my comments. The manuscript has improved. English has improved however there are still several mistakes (spell check and style) that should be amended before the publication.
Reviewer 3 Report
Tha authors addressed all the raised items.